# A cost-effectiveness analysis of three surgical options for treating displaced femoral neck fractures in active older patients in Japan: A full economic evaluation

**Kazutaka Yokoyama**[1]*, **Yoko Akune**[1], **Hiroyuki Katoh**[2], **Seiji Bito**[3], **Yoshinari Fujita**[4], **Rei Goto**[5], **Keita Yamauchi**[1]

1 Graduate School of Health Management, Keio University, Kanagawa, Japan, 2 Department of Orthopaedic Surgery, Surgical Science, Tokai University School of Medicine, Kanagawa, Japan, 3 Department of Clinical Epidemiology, National Hospital Organization Tokyo Medical Center, Tokyo, Japan, 4 Department of Orthopaedic Surgery, National Hospital Organization Tokyo Medical Center, Tokyo, Japan, 5 Graduate School of Business Administration, Keio University, Kanagawa, Japan

* h19831ky@keio.jp

**Data Availability Statement:** All relevant data are within the manuscript and its Supporting information files.

## Abstract

For older patients with displaced femoral neck fractures, in which primary osteosynthesis is usually not indicated, there are three primary prosthetic options—bipolar hemiarthroplasty (BHA), single-bearing total hip arthroplasty (SB-THA), and dual-mobility THA (DM-THA). However, the optimal choice for managing displaced femoral neck fractures remains controversial. Accordingly, this study aimed to evaluate the cost-effectiveness of BHA, SB-THA, and DM-THA in active older patients with displaced femoral neck fractures. A decision tree combined with a Markov model was employed to analyze the cost and quality-adjusted life years (QALYs) of BHA, SB-THA, and DM-THA for the management in the Japanese healthcare system. By simulating the five-year trajectory of a 75-year-old woman treated for a displaced femoral neck fracture, the cost-effectiveness of the three surgical options was evaluated. One-way sensitivity analysis and probabilistic sensitivity analysis (PSA) were used to assess parameter uncertainty. Additionally, two scenario analyses were conducted for other settings. The treatment was considered to be cost-effective when the incremental cost-effectiveness ratio (ICER) was below the 5,000,000 yen/QALY threshold. Compared with BHA, SB-THA exhibited higher costs but greater health benefits, resulting in an ICER of 1,499,440 yen/QALY. DM-THA offered additional health benefits compared with SB-THA, with an ICER of 4,145,777 yen/QALY. One-way sensitivity analysis revealed some influential parameters. PSA indicated that the probability of DM-THA, SB-THA, and BHA being cost-effective was 40.1%, 38.5%, and 21.4%, respectively. SB-THA was more cost-effective than BHA in patients aged 65–85 years, while DM-THA was more cost-effective than SB-THA in patients aged 65–75 years. The results suggest that SB-THA is a cost-effective alternative to BHA for displaced femoral neck fractures in active older patients, whereas DM-THA is more cost-effective than SB-THA in relatively younger patients. It is, therefore, recommended that orthopedic surgeons

**Funding:** This study was supported by JST SPRING (grant number, JPMJSP2123) and the Keio University Doctorate Student Grant-in-Aid Program of the Ushioda Memorial Fund. The funders had no role in the study design, data collection and analysis, decision to publish, or preparation of the manuscript.

**Competing interests:** The authors have declared that no competing interests exist.

select the most appropriate surgical option based on the individual patient's physiological age.

## Introduction

Hip fractures pose significant risks such as health complications, prolonged hospitalization, and reduced health-related quality of life in older patients [1–3]. These risks have been affecting both patients and their families for a long time, presenting serious public health problems worldwide. According to a new global epidemiological study, the number of hip fractures is estimated to nearly double worldwide from 2018 to 2050 [4]. This number has also increased in Japan, from 76,600 in 1992 to 193,400 in 2017 [5], with a projected increase to 290,000 by 2030 [6].

Considering that primary osteosynthesis is usually not indicated, there are three primary prosthetic options generally available for older patients with displaced femoral neck fractures —bipolar hemiarthroplasty (BHA), conventional single-bearing total hip arthroplasty (SB-THA), and dual-mobility THA (DM-THA), which is the most recent surgical option [7, 8]. A randomized controlled trial (HEALTH trial) showed no significant difference between THA and hemiarthroplasty (HA) in terms of the risk of unplanned secondary hip procedures over a period of two years in patients with displaced femoral neck fractures [9], while the risk of dislocation was higher in the THA group than in the HA group [10]. Recently, DM-THA implants have been introduced to reduce the risk of dislocation [11], but intermediate- and long-term data on these surgical options are lacking. In Japan, only a few centers with advanced surgical techniques offer THA as the first choice for femoral neck fractures.

The optimal choice for managing displaced femoral neck fractures remains controversial. Japan became the country with the longest life expectancy in the world in 1985 and has since maintained this status. Since hip fractures are common injuries among older adults, it is important to conduct economic evaluations of the surgical options using Japanese data. However, to date, no study has reported on the cost-effectiveness of the three surgical options for displaced femoral neck fractures using Japanese data.

Therefore, in the present study, we hypothesized that DM-THA might be a cost-effective option for the active older adults, even though it is the most expensive of the three surgical options. This study aimed to clarify the cost-effectiveness of BHA, SB-THA, and DM-THA and to ascertain whether the higher cost of DM-THA is justified by its clinical benefits over SB-THA and BHA in patients with displaced femoral neck fractures.

## Materials and methods

### Ethics statement

All data used in this study were collected from books, articles, and open access data from the Japanese Ministry of Health, Labour and Welfare. Because no individual patient data were used, this study did not require patient consent or institutional review board approval.

### Study design

A decision tree model combined with a Markov model was built using TreeAge Pro Healthcare Version 2024 (Build-Id: 24.2.0-v20240709; TreeAge Software, LLC, Williamstown, MA, United States [US]) to evaluate the cost-effectiveness of three surgical options, conduct one-

way sensitivity analyses, and perform probabilistic sensitivity analyses (PSAs) (Fig 1). The model was designed and built from the Japanese public healthcare payer's perspective, and effectiveness was measured using quality-adjusted life years (QALYs). QALYs are calculated by multiplying "quality of life" by "life years." We examined the costs (Japanese yen [¥]) and QALYs associated with BHA, SB-THA, and DM-THA for the treatment of displaced femoral neck fractures and compared BHA, SB-THA, and DM-THA. Other strategies considered in cases with non-displaced fractures, such as conservative treatment and internal fixation, were not considered in this study.

The base case for the model was set to be a 75-year-old woman with a displaced femoral neck fracture and no contraindications for surgery. A 75-year-old woman was chosen as the model base case because femoral neck fractures occur more frequently in women than in men and the age of 75 years is close to the mean age reported in a systematic review and meta-analysis [12]. Scenario analyses were conducted for other age groups and male patients. Patients with displaced femoral neck fractures were allocated to either BHA, SB-THA, or DM-THA and then assumed to transition into one of the following modeled health states: well

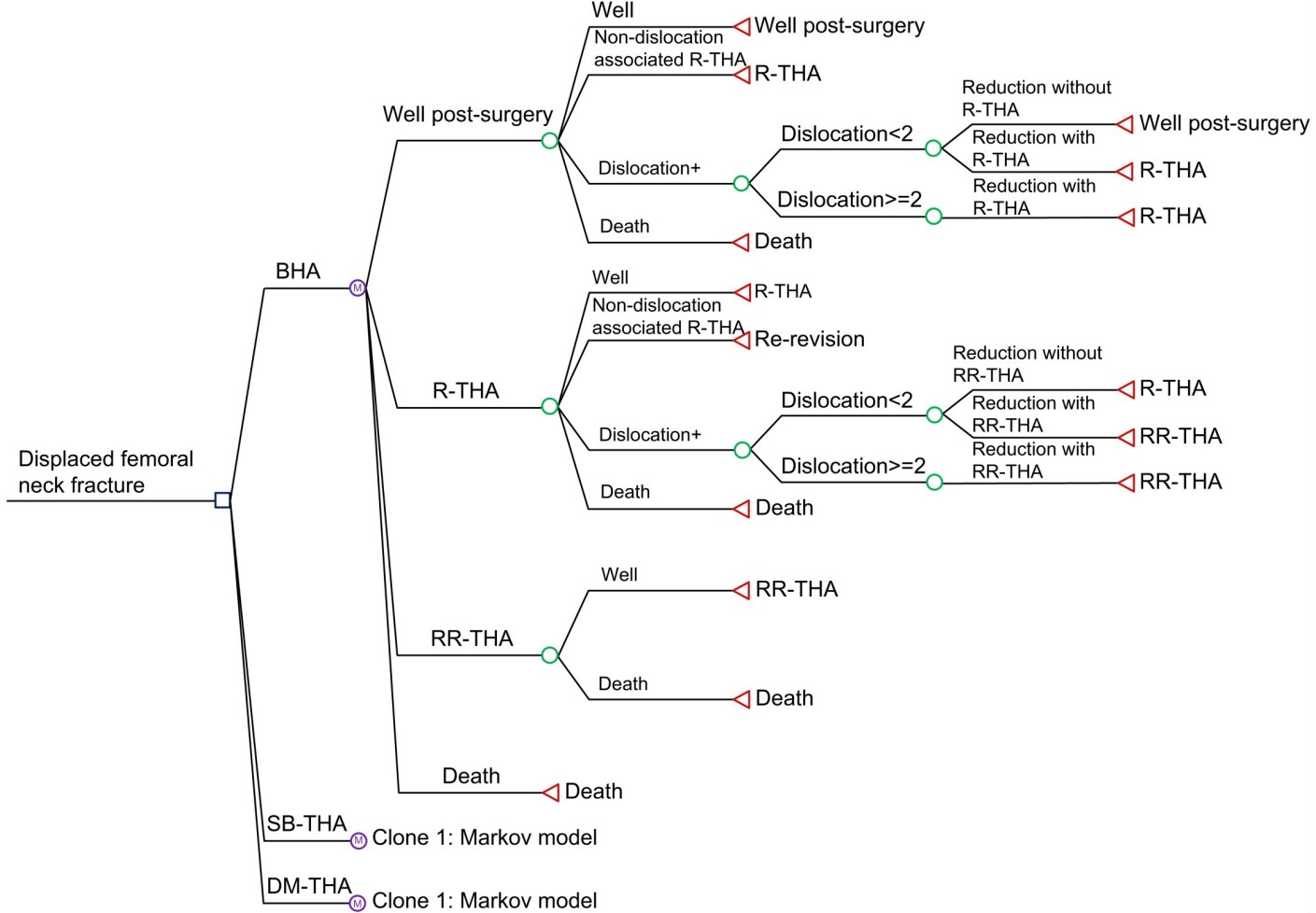

**Fig 1. Markov model used in the decision tree model.** While the tree is expanded only for BHA, identical Markov models are also used for SB-THA and DM-THA. BHA, bipolar hemiarthroplasty; DM-THA, dual-mobility total hip arthroplasty; R-THA, revision total hip arthroplasty; RR-THA, re-revision total hip arthroplasty; SB-THA, single-bearing total hip arthroplasty.

post-surgery, dislocation, revision THA (R-THA), re-revision THA (RR-THA), and death. In this model, we mainly focused on "dislocation" because it may be the key factor for quality of life. However, we did not include the "infection" state in the model because there was no significant difference between THA and HA with regard to the risk of infection [3, 10]. Transitions were made on an annual cycle basis in the Markov model. An annual cycle was considered to represent a realistic clinical period during which revisions are prone to occur based on previous studies [13, 14].

A first-order Monte Carlo simulation (microsimulation) was conducted to evaluate stochastic uncertainty, and a tracker variable was used to consider the occurrence of a dislocation. The number of trials for the microsimulation was set to 100,000. The model assumes that the first dislocation has a possibility to be reduced by closed reduction and to be successfully managed. The second dislocation is assumed to require revision surgery, in which a DM-THA implant is used. In cases where a second revision (RR-THA) is deemed necessary, it is assumed, based on a previous study, that the procedure is successful and requires no further revisions thereafter [14]. At all times, patients are considered to be at risk of death. Death was considered an absorbing state because no further transitions can occur once a patient has entered this health state.

The time horizon for this analysis was set to five years. Both the cost and effectiveness were discounted at a rate of 2.0% per year in accordance with the Guideline for Preparing Cost-Effectiveness Evaluation to the Central Social Insurance Medical Council [15]. The age-specific mortality rate and life expectancy were obtained from the data published by the Japanese Ministry of Health, Labour and Welfare in the abridged life table of Japan 2022 [16, 17] and are summarized in S1 and S2 Tables. In this study, an "active older patient" is defined as a self-sufficient, physically active person aged 65 years or older whose mortality rate is similar to that provided in the abridged life table of Japan.

The primary outcomes were the cost (in Japanese yen) and effectiveness (in QALY) of the three strategies. The incremental cost-effectiveness ratio (ICER) was estimated to evaluate cost-effectiveness. An ICER is calculated by dividing the "incremental costs" by the "incremental effectiveness." The results of the ICER were compared with the established willingness-to-pay (WTP) thresholds for cost-effectiveness. A WTP threshold is a value that estimates how much a patient is willing to pay for a health benefit. In general, the Japanese Ministry of Health, Labour and Welfare considers interventions with an estimated ICER of <5,000,000 yen/QALY to be cost-effective. To ensure that the model would not deviate from normal practice, an orthopedic surgeon [YF] evaluated the clinical assumptions, and two health economists [RG and YA] evaluated the methodology and results.

## Model inputs

The annual transition probabilities by age group are presented in Table 1. The probabilities of dislocation after SB-THA for specific age groups were derived from a previous study [18], and

**Table 1. Annual transition probabilities by age group.**

| Annual transition probabilities | 65–69 years | 70–74 years | 75–79 years | 80–84 years | 85–89 years |
|---|---|---|---|---|---|
| Probability of dislocation after SB-THA [18] | 0.0240 | 0.0235 | 0.0215 | 0.0190 | 0.0132 |
| Probability of revision dislocation after SB-THA [18, 19] | 0.0097 | 0.0103 | 0.0086 | 0.0067 | 0.0040 |
| Probability of revision non-dislocation after SB-THA [18, 19] | 0.0211 | 0.0225 | 0.0187 | 0.0146 | 0.0087 |

SB-THA, single-bearing total hip arthroplasty.

dislocations were managed by either closed or open reduction. The probabilities of revision due to dislocation after SB-THA and those due to causes other than dislocation after SB-THA were calculated using data from two previous studies [18, 19]. Other parameters such as clinical effectiveness, disutility, health state utility, and cost data are listed in Table 2 and detailed in S3 Table, which is a summarized table of data published by the Japanese Health Insurance Federation for Surgery [20].

**Table 2. Parameters for the decision model.**

| Parameter description | Base case estimate | For sensitivity analysis | |
|---|---|---|---|
| | Mean | Low | High |
| *Clinical effectiveness data* | | | |
| Relative risk of dislocation of DM-THA compared with SB-THA [21] | 0.17 | 0.04 | 0.79 |
| Relative risk of dislocation of DM-THA compared with BHA [21] | 0.41 | 0.19 | 0.87 |
| Relative risk of revision dislocation of BHA compared with DM-THA [22] | 7.16[a] | 2.18[a] | 23.58[a] |
| Relative risk of revision non-dislocation of BHA compared with DM-THA [22] | 0.87[a] | 0.40[a] | 1.88[a] |
| Relative risk of revision dislocation of DM-THA compared with SB-THA [23] | 0.40[a] | 0.27[a] | 0.59[a] |
| Relative risk of revision non-dislocation of DM-THA compared with SB-THA [23] | 0.93[a] | 0.75[a] | 1.16[a] |
| *Disutility data* | | | |
| Disutility due to dislocation [14] | -0.110 | -0.153 | -0.067 |
| Disutility due to R-THA [13] | -0.185 | -0.258 | -0.112 |
| Disutility due to RR-THA [13] | -0.287 | -0.400 | -0.174 |
| *Health state utility data* | | | |
| Health state utility of well post-BHA [24] | 0.820 | 0.660 | 0.980 |
| Health state utility of well post-SB-THA [24] | 0.870[a] | 0.720[a] | 1.000[a] |
| Health state utility of well post-DM-THA [24] | 0.870 | 0.720 | 1.000 |
| Health state utility of well post-R-THA [13, 24] | 0.685[a] | 0.535[a] | 0.835[a] |
| Health state utility of well RR-THA [13, 24] | 0.583[a] | 0.433[a] | 0.733[a] |
| Health state utility of mortality [14] | 0.000 | 0.000 | 0.000 |
| *Cost data (yen)* | | | |
| Costs after dislocation [20] | 220,679[a] | 134,173[a] | 307,185[a] |
| Cost of BHA implants [25] | 572,000[a] | 572,000[a] | 572,000[a] |
| Cost of SB-THA implants [25] | 611,000[a] | 611,000[a] | 611,000[a] |
| Cost of DM-THA implants [25] | 714,000[a] | 714,000[a] | 714,000[a] |
| Fee for medical service of BHA [20] | 195,000 | 195,000 | 195,000 |
| Fee for medical service of SB-THA [20] | 376,900 | 376,900 | 376,900 |
| Fee for medical service of DM-THA [20] | 376,900 | 376,900 | 376,900 |
| Fee for medical service of R-THA [20] | 548,100 | 548,100 | 548,100 |
| Initial year costs of BHA [26] | 2,122,628[a] | 1,290,558[a] | 2,954,698[a] |
| Initial year costs of THA [26] | 2,191,285[a] | 1,332,301[a] | 3,050,269[a] |
| Costs of well post-surgery [27] | 1,399,413[a] | 1,052,656 | 1,746,170 |

[a] Assumption or calculated values: refer to S3 Table.

When only the point estimate was known, the range of variation was set as follows: "standard error" = "point estimate" × 0.2, "Low" = "point estimate"–"standard error" × 1.96, "High" = "point estimate" + "standard error" × 1.96.

The revision rate of the medical payment system was utilized to standardize the data concerning the "initial year costs of BHA," "initial year costs of THA," and "costs of well post-surgery" to that of the year 2022.

BHA, bipolar hemiarthroplasty; DM-THA, dual-mobility total hip arthroplasty; R-THA, revision total hip arthroplasty; RR-THA, re-revision total hip arthroplasty; SB-THA, single-bearing total hip arthroplasty.

## Sensitivity analysis

One-way sensitivity analyses were performed to assess uncertainty and robustness. The tornado diagram summarizes the results of the one-way sensitivity analyses. The range of variation was adopted from available previous studies. When only the point estimate was available, the range of variation was set as follows: "standard error" = "point estimate" × 0.2, "Low" = "point estimate"–"standard error" × 1.96, "High" = "point estimate" + "standard error" × 1.96. In accordance with the Japanese guidelines, the discount rate was subjected to a one-way sensitivity analysis and was changed at the same rate of 0.0–4.0% per year for both cost and effectiveness [15].

PSA was used to assess uncertainty and evaluate the probability of cost-effectiveness. For the PSA, gamma, log-normal, normal, and beta distributions were used for costs, relative risks, disutilities, and utilities and probabilities, respectively. In the analysis using TreeAge Pro Healthcare, the number of trials for microsimulation was set to 100,000, and the number of samples for the PSA was set to 10,000. For the incremental cost-effectiveness scatter plot, the range of iterations was set from 1 to 10,000.

## Scenario analysis

Two scenario analyses (scenario analysis A and scenario analysis B) were performed. First, in addition to the 75-year-old female patient, scenario analyses were conducted for patients of the other age groups and male patients. Specifically, the age range varied from 65 years to 85 years. Second, to assess shorter periods, the time horizon was changed from five years to four years, three years, two years, and one year.

## CHEERS guidelines

This study was performed in accordance with the Consolidated Health Economic Evaluation Reporting Standards (CHEERS) 2022 checklist (see S1 File).

## Results

Fig 2 and Table 3 present the microsimulation results of the model base case of a 75-year-old female patient followed over a period of five years. The cost was 8,326,071 yen, and the QALYs were 3.792 for the BHA strategy; these figures were 8,669,497 yen and 4.021, respectively, for the SB-THA strategy. In other words, SB-THA was more costly than BHA but resulted in more health benefits (measured in QALYs). To explain in more detail, the estimated incremental cost per patient was 343,426 yen, whereas the estimated incremental QALYs were 0.229, resulting in a cost per QALY of 1,499,440 yen/QALY. Similarly, DM-THA was more costly than SB-THA but resulted in more health benefits. The estimated incremental cost per patient was 53,904 yen, whereas the estimated incremental QALYs were 0.013, resulting in a cost per QALY of 4,145,777 yen/QALY.

Tornado diagrams were generated by changing the parameters of the model base case group. Looking at the ICER for SB-THA compared with that for BHA (Fig 3A), there are two infinite signs and two bars showing threshold values. An infinity sign means that the ICER ("incremental cost"/ "incremental effectiveness") becomes undefined because the "incremental effectiveness" approaches zero. If "health state utility of well post-BHA" is better than "health state utility of well post-SB-THA," SB-THA would be "dominated." If a surgical option is higher in cost but equivalent or lower in effectiveness relative to the comparator, the surgical option is called "dominated." As for the two bars showing the threshold values, specific values of upper ICER and lower ICER are described. If the initial year cost of SB-THA increases to

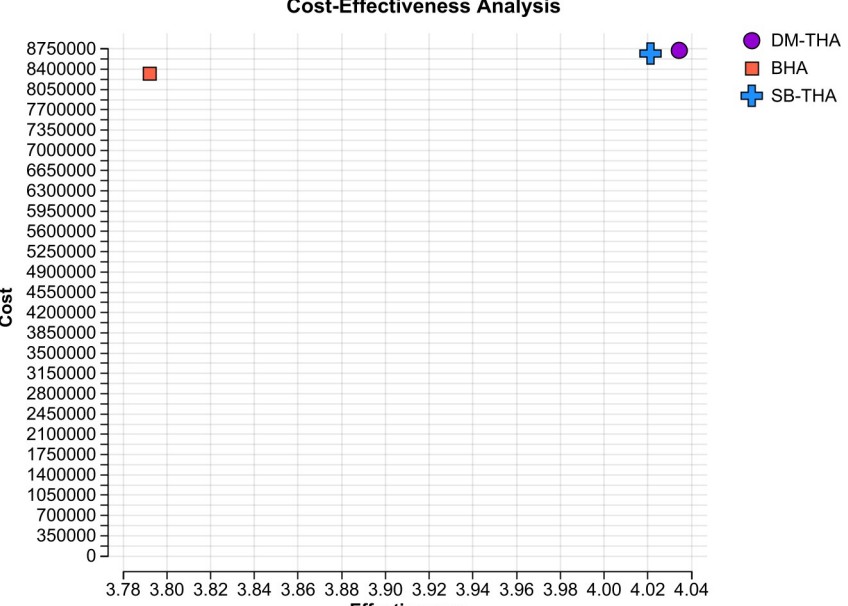

**Fig 2. ICER results of female patients aged 75 years over a five-year period.** BHA, bipolar hemiarthroplasty; DM-THA, dual-mobility total hip arthroplasty; ICER, incremental cost-effectiveness ratio; SB-THA, single-bearing total hip arthroplasty.

2,983,027 yen, with the other parameters remaining the same, SB-THA would no longer be cost-effective (exceeding the 5,000,000 yen/QALY WTP threshold). Similarly, if the initial year cost of BHA decreases to 1,320,872 yen, with the other parameters remaining the same, SB-THA would no longer be cost-effective. Further, looking at the ICER for DM-THA compared with that for SB-THA (Fig 3B), there are two infinite signs and four bars showing threshold values. If "health state utility of well post-SB-THA" is better than "health state utility of well post-DM-THA," DM-THA would be "dominated." In addition, if parameters such as the "relative risk of revision non-dislocation of DM-THA compared with SB-THA," "relative risk of dislocation of DM-THA compared with SB-THA," "disutility due to dislocation," and "health state utility of well post-R-THA" increases to 0.958, 0.348, -0.081, and 0.83, respectively, DM-THA would no longer be cost-effective (exceeding the 5,000,000 yen/QALY WTP threshold). To explain further, for DM-THA to be cost-effective, the relative risk of dislocation should be less than 0.348, meaning that the number of dislocation events after DM-THA would need to be reduced by at least 65.2% compared with those after SB-THA.

**Table 3. Data table for Fig 2.**

| Intervention | Cost (yen) | Incremental cost (yen) | QALYs | Incremental QALYs | ICER (yen/QALY) |
|---|---|---|---|---|---|
| BHA | 8,326,071 | NA | 3.792 | NA | NA |
| SB-THA | 8,669,497 | 343,426 | 4.021 | 0.229 | 1,499,440 |
| DM-THA | 8,723,401 | 53,904 | 4.034 | 0.013 | 4,145,777 |

BHA, bipolar hemiarthroplasty; DM-THA, dual-mobility total hip arthroplasty; ICER, incremental cost-effectiveness ratio; NA, not applicable; QALY, Quality-adjusted life year; SB-THA, single-bearing total hip arthroplasty.

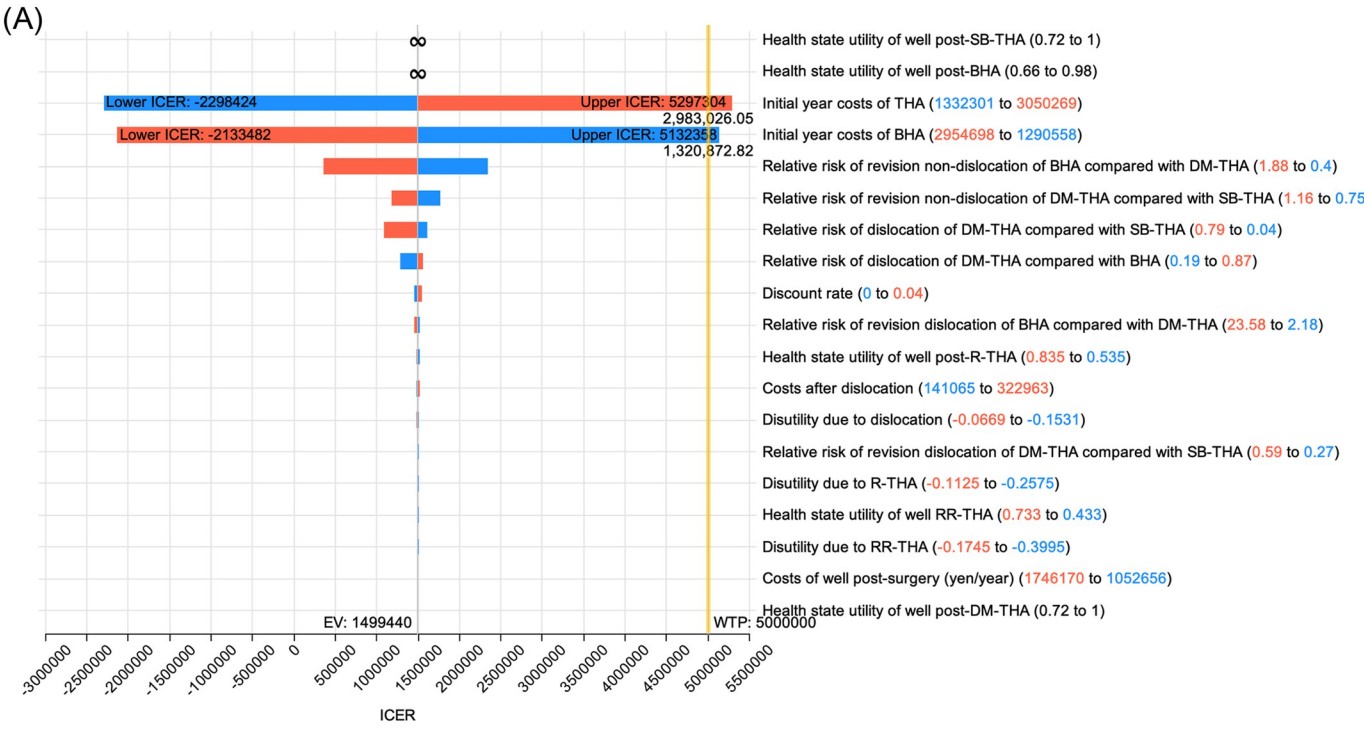

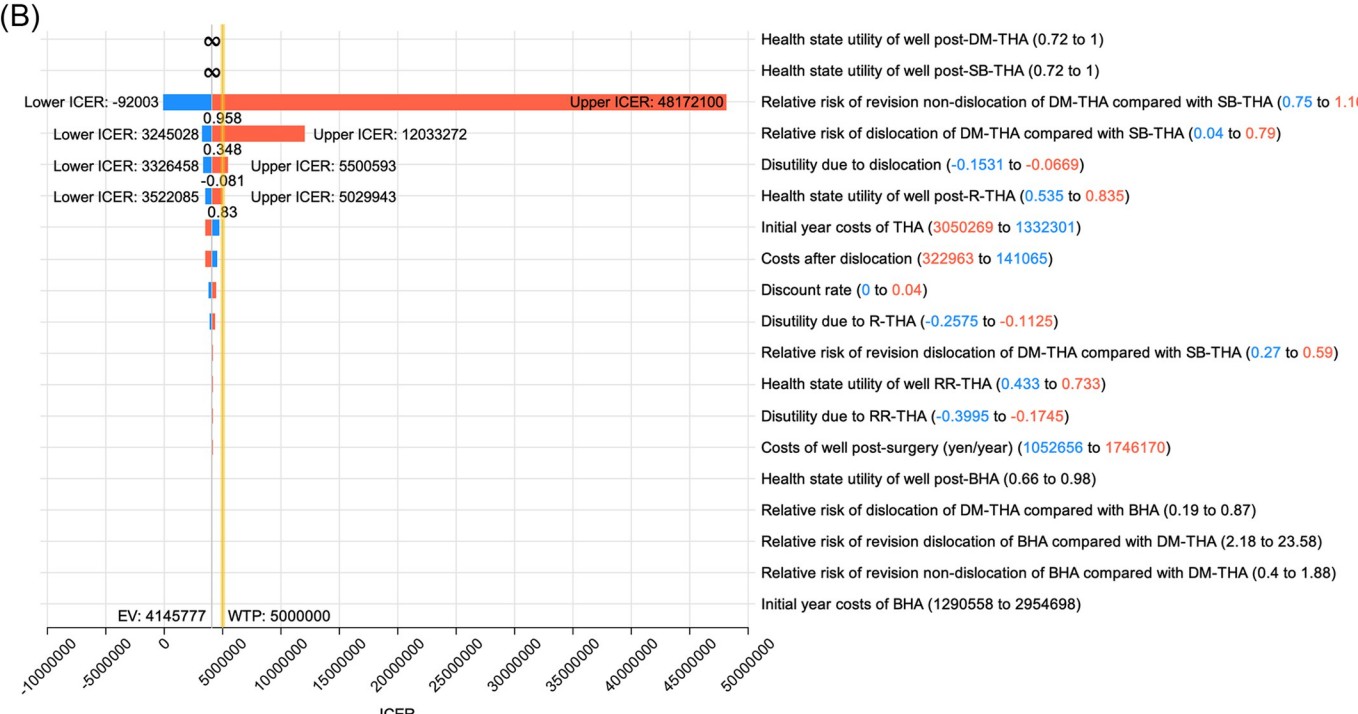

**Fig 3. Tornado diagram of the ICER.** A: SB-THA versus BHA. B: DM-THA versus SB-THA. An infinity sign means that the ICER becomes undefined because the incremental effectiveness approaches zero. Bars crossing the WTP threshold show threshold values, upper ICER, and lower ICER. The yellow line shows the WTP threshold of 5,000,000 yen/QALY. BHA, bipolar hemiarthroplasty; DM-THA, dual-mobility total hip arthroplasty; EV, expected value; ICER, incremental cost-effectiveness ratio; R-THA, revision total hip arthroplasty; RR-THA, re-revision total hip arthroplasty; RR, relative risk; SB-THA, single-bearing total hip arthroplasty; WTP, willingness-to-pay.

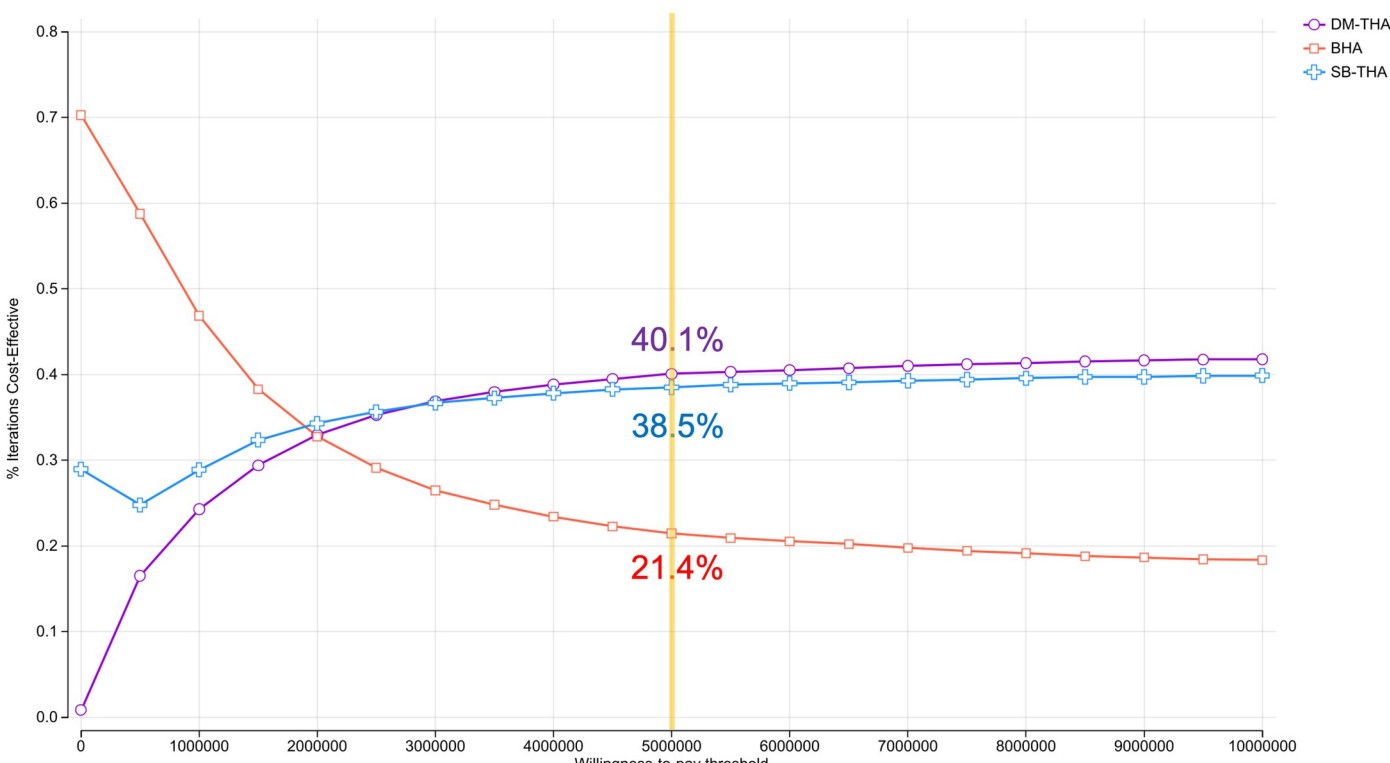

**Fig 4. Cost-effectiveness acceptability curves of the three surgical options.** The probability that the surgical option is cost-effective at differing willingness-to-pay values for BHA, SB-THA, and DM-THA in female patients aged 75 years. The color of the line denotes the surgical options: red line, BHA; blue line, SB-THA; and violet line, DM-THA. The yellow line shows the willingness-to-pay threshold of 5,000,000 yen/QALY. BHA, bipolar hemiarthroplasty; DM-THA, dual-mobility total hip arthroplasty; SB-THA, single-bearing total hip arthroplasty.

PSA was conducted to assess uncertainty. The incremental cost-effectiveness scatter plot showed uncertainty in this model as the points were distributed across all four quadrants (S1 and S2 Figs, S4 and S5 Tables). According to the cost-effectiveness acceptability curves of the three surgical options, the probabilities of DM-THA, SB-THA, and BHA being cost-effective were estimated to be 40.1%, 38.5%, and 21.4%, respectively, at the WTP threshold of 5,000,000 yen/QALY for the model case (Fig 4).

As in scenario analysis A, microsimulations were conducted for the other age groups and for male patients (Table 4). SB-THA was more cost-effective than BHA for patients aged 65–85 years. In addition, DM-THA was more cost-effective than SB-THA for patients aged 65–75 years at the WTP threshold of 5,000,000 yen/QALY. However, DM-THA was not cost-effective for patients aged 80 years and older in Japan.

As in scenario analysis B, the time horizon of the model was changed from five years to four years, three years, two years, and one year (Table 5). SB-THA was more cost-effective than BHA for patients with the time horizon from two to five years. In contrast, DM-THA was more cost-effective than SB-THA for patients only with the five-year time horizon.

## Discussion

In this study, we conducted a full economic evaluation of three surgical interventions—BHA, SB-THA, and DM-THA—for displaced femoral neck fractures in active older patients in

**Table 4. ICER results of scenario analysis A (Japanese yen) stratified by age and sex over a five-year period.**

| BHA versus SB-THA | | |
| --- | --- | --- |
| **Age, years** | **Five-year period** | |
| | **Female** | **Male** |
| 65 | ¥1,518,334 | ¥1,544,124 |
| 70 | ¥1,538,658 | ¥1,561,974 |
| 75 | ¥1,499,440 | ¥1,540,972 |
| 80 | ¥1,465,529 | ¥1,526,412 |
| 85 | ¥1,425,169 | ¥1,534,205 |
| **SB-THA versus DM-THA** | | |
| **Age, years** | **Five-year period** | |
| | **Female** | **Male** |
| 65 | ¥3,039,119 | ¥3,081,576 |
| 70 | ¥3,006,061 | ¥3,458,102 |
| 75 | ¥4,145,777 | ¥4,529,640 |
| 80 | ¥5,897,579 | ¥6,456,597 |
| 85 | ¥12,170,692 | ¥13,378,666 |

BHA, bipolar hemiarthroplasty; DM-THA, dual-mobility total hip arthroplasty; ICER, incremental cost-effectiveness ratio; SB-THA, single-bearing total hip arthroplasty.

Japan, addressing an important gap in the literature. A full economic evaluation is a type of health economic analysis that compares the costs and effectiveness of multiple interventions. Compared with BHA, both SB-THA and DM-THA are considered cost-effective for a 75-year-old woman with a displaced femoral neck fracture because they clear the 5,000,000 yen/QALY threshold. The findings of this study suggest that SB-THA could be a cost-effective alternative to BHA and that DM-THA might be more cost-effective than SB-THA for relatively younger age groups.

Multicenter randomized controlled trials, such as the DUALITY and DISTINCT trials, are currently underway to clarify the risk of hip dislocation after DM-THA [28, 29]. If the relative risk of dislocation turns out to be higher than that reported in previous studies, the economic evaluation of DM-THA implants should be reevaluated.

The thresholds for cost-effectiveness are set for each country: 5,000,000 yen per QALY for Japan, 50,000 US dollars (US$) per QALY for the US, and 20,000–30,000 pounds sterling (£) per QALY for the United Kingdom [30]. The ICER results of scenario analysis A stratified by age and sex over a five-year period were converted to £ and US$ for reference (Table 4 and

**Table 5. ICER results of scenario analysis B (Japanese yen) stratified by time horizon.**

| Time horizon | BHA versus SB-THA | SB-THA versus DM-THA |
| --- | --- | --- |
| One year | ¥5,807,625 | ¥45,279,662 |
| Two years | ¥3,090,964 | ¥19,375,309 |
| Three years | ¥2,197,193 | ¥10,783,733 |
| Four years | ¥1,756,807 | ¥6,652,654 |
| Five years | ¥1,499,440 | ¥4,145,777 |

BHA, bipolar hemiarthroplasty; DM-THA, dual-mobility total hip arthroplasty; ICER, incremental cost-effectiveness ratio; SB-THA, single-bearing total hip arthroplasty.

S6 Table) using the average currency conversion rate of the 2022 fiscal year (¥162 is equivalent to £1 and ¥131 is equivalent to US$1) [31]. Since the model's costs are specific to Japan, generalizing the results to other countries is not feasible, and the cost-effective boundary for age might differ depending on the ICER threshold and currency conversion rate in each country.

In scenario analysis B (Table 5), SB-THA and DM-THA are not cost-effective with a one-year time horizon. This is partly because the cost of implants and the initial year cost are expensive. SB-THA is more cost-effective than BHA with a two- to five-year time horizon thanks to the incremental QALYs. DM-THA is more cost-effective than SB-THA only with a five-year time horizon partly because the relative risk of dislocation of DM-THA compared with SB-THA is estimated to be 0.17 (Table 2).

Our study has several strengths. First, the comprehensive methodology of the study, employing a decision tree and Markov model combined with sensitivity analyses, enhanced the reliability and applicability of the results. Second, the consideration of different age groups and sexes in the scenario analysis A reflects a thorough approach that acknowledges the diversity of patient populations. We have presented the results for the 75-year-old female group, which represents the majority of the patient population. In addition, the scenario analyses present the results for patients of other age groups and male patients. Third, this study assessed short- and middle-term cost-effectiveness in scenario analysis B. The time horizon was originally set to five years, but we changed the time horizon from five years to four years, three years, two years, and one year. Changing the time horizon broadened the relevance of the study, making it a valuable resource for orthopedic surgeons and healthcare decision-makers. Fourth, to the best of our knowledge, this is the first study to examine the cost-effectiveness of three surgical options for treating displaced femoral neck fractures.

This study also has some limitations. First, the longest time horizon of this study was limited to five years, which may not fully capture the long-term effectiveness and cost associated with these surgical options. Long-term follow-up studies would be beneficial for understanding the sustained effects of these interventions. However, if a surgical option is cost-effective over a five-year period, it is likely to remain cost-effective or even cost-saving (a dominant strategy) over a longer time horizon, as suggested by a previous study [14]. Additionally, given the five-year mortality rate after a hip fracture in Japan is reported to be 45.6% [32], a five-year time horizon might be considered acceptable. Second, in this study, we assumed that all second revisions (RR-THAs) would be successful, without considering the possibility of further complications or surgeries. This assumption may underestimate the long-term costs and effectiveness of the surgical options, particularly DM-THA. Third, the PSA revealed a degree of uncertainty in the cost-effectiveness of DM-THA. This uncertainty highlights the need for further research, particularly in regard to the relative risks of dislocation and revision associated with DM-THA. Fourth, data regarding "costs after dislocation" are lacking in Japan. These costs were assumed as the average costs of "closed reduction for hip dislocation" and "open reduction for hip dislocation" (S3 Table) calculated to be 220,679 yen, which is similar to those reported in the United Kingdom and Canada [13, 14]. Moreover the "costs after dislocation" was not an influential key parameter in one-way sensitivity analyses. Registry data regarding these costs are not available in Japan, so further studies using Japanese registry data are needed to address this gap, presenting a challenge for future studies. Fifth, the study did not explicitly discuss the waiting time for surgery, dementia in patients, skill of orthopedic surgeons, surgical approaches such as the posterior or direct anterior approaches, and use of cement, even though these factors can affect the outcomes [11, 33, 34]. This is partly because studies regarding the effects of these factors are insufficient. More complex models can be developed if real-world clinical studies investigating these factors are published in the future. Sixth, the study does not consider the patient's condition or decision-making process involved in treating a

displaced femoral neck fracture. However, it does contribute to our understanding of cost-effectiveness, which can be useful in guiding the decisions made by both the patient and the doctor.

Although this study provides valuable insights into the cost-effectiveness of surgical options for displaced femoral neck fractures in Japan, its generalizability to other healthcare settings, particularly to those with different healthcare systems and cost structures, is limited. Ethical considerations, such as the potential impact of cost-driven decisions on the quality of patient care, should also be considered when applying these findings to clinical practice.

## Conclusion

The findings of this study suggest that SB-THA is a more cost-effective option than BHA for patients aged 65–85 years, while DM-THA is a more cost-effective option than SB-THA for patients aged 65–75 years over a five-year period. For 75-year-old female patients, our results suggest that SB-THA becomes more cost-effective than BHA after two years, and DM-THA becomes more cost-effective than SB-THA after five years. Orthopedic surgeons should select the most appropriate surgical option by considering the patient's expected mortality and overall health condition.

## Supporting information

**S1 Table. Abridged life table of Japan, 2022 (male).**
(DOCX)

**S2 Table. Abridged life table of Japan, 2022 (female).**
(DOCX)

**S3 Table. Costs of the currently available surgeries for hip dislocation.** "Costs after dislocation" in Table 2 is considered as the average costs of "closed reduction for hip dislocation" and "open reduction for hip dislocation": (42,738+398,620)/2.
(DOCX)

**S4 Table. Data table of S1 Fig.** IC, incremental cost; ICER, incremental cost-effectiveness ratio; IE, incremental effectiveness; QALY, Quality-adjusted life year.
(DOCX)

**S5 Table. Data table of S2 Fig.** IC, incremental cost; ICER, incremental cost-effectiveness ratio; IE, incremental effectiveness; QALY, Quality-adjusted life year.
(DOCX)

**S6 Table. ICER results of scenario analysis A (Japanese yen, pound sterling, and US dollar) stratified by age and sex over a five-year period.** The values in pounds sterling (£) and US dollars (US$) were calculated based on the average currency conversion rates in 2022: ¥162 = £1 and ¥131 = US$1. BHA, bipolar hemiarthroplasty; DM-THA, dual-mobility total hip arthroplasty; ICER, incremental cost-effectiveness ratio; SB-THA, single-bearing total hip arthroplasty.
(DOCX)

**S1 Fig. Incremental cost-effectiveness scatter plot of SB-THA and BHA.** The scatter plot shows outcomes of SB-THA versus BHA in a 75-year-old female patient. Each point represents a single simulated result of 10,000 simulations. The green ellipse denotes the 95% confidence ellipse. BHA, bipolar hemiarthroplasty; SB-THA, single-bearing total hip arthroplasty; WTP,

willingness-to-pay.
(TIF)

**S2 Fig. Incremental cost-effectiveness scatter plot of DM-THA and SB-THA.** The scatter plot shows cost outcomes of DM-THA versus SB-THA in a 75-year-old female patient. Each point represents a single simulated result of 10,000 simulations. The green ellipse denotes the 95% confidence ellipse. DM-THA, dual-mobility total hip arthroplasty; SB-THA, single-bearing total hip arthroplasty; WTP, willingness-to-pay.
(TIF)

**S1 File. CHEERS 2022 checklist.** *From*: Husereau D, Drummond M, Augustovski F, de Bekker-Grob E, Briggs AH, Carswell C, et al. Consolidated Health Economic Evaluation Reporting Standards (CHEERS) 2022 Explanation and Elaboration: A Report of the ISPOR CHEERS II Good Practices Task Force. Value Health. 2022;25(1):10–31. https://doi.org/10.1016/j.jval.2021.10.008.
(DOCX)

## Acknowledgments

We would like to acknowledge the useful advice provided by the reviewers of PLOS ONE.

## Author Contributions

**Conceptualization:** Kazutaka Yokoyama.

**Data curation:** Kazutaka Yokoyama.

**Formal analysis:** Kazutaka Yokoyama.

**Funding acquisition:** Kazutaka Yokoyama.

**Investigation:** Kazutaka Yokoyama.

**Methodology:** Kazutaka Yokoyama, Yoko Akune, Rei Goto.

**Project administration:** Kazutaka Yokoyama.

**Resources:** Kazutaka Yokoyama.

**Software:** Kazutaka Yokoyama.

**Supervision:** Kazutaka Yokoyama, Yoko Akune, Hiroyuki Katoh, Seiji Bito, Yoshinari Fujita, Rei Goto, Keita Yamauchi.

**Validation:** Kazutaka Yokoyama, Yoko Akune, Rei Goto.

**Visualization:** Kazutaka Yokoyama.

**Writing – original draft:** Kazutaka Yokoyama.

**Writing – review & editing:** Kazutaka Yokoyama, Yoko Akune, Hiroyuki Katoh, Seiji Bito, Yoshinari Fujita, Rei Goto, Keita Yamauchi.

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
