## [Decision Letter · Decision Letter 0]

24 Jul 2024

PONE-D-24-17748A cost-effectiveness analysis of three surgical options for treating displaced femoral neck fractures in active older patients in Japan: a full economic evaluationPLOS ONE

Dear Dr. Yokoyama,

Thank you for submitting your manuscript to PLOS ONE. After careful consideration, we feel that it has merit but does not fully meet PLOS ONE’s publication criteria as it currently stands. Therefore, we invite you to submit a revised version of the manuscript that addresses the points raised during the review process.

Your manuscript is well written, the topic is interesting in all over the world. Please address all reviewers comments to improve your paper.

We look forward to receiving your revised manuscript.

Kind regards,

Hans-Peter Simmen, M.D., Professor of Surgery

Academic Editor

PLOS ONE

Additional Editor Comments (if provided):

Reviewers' comments:

Reviewer's Responses to Questions

**Comments to the Author**

1. Is the manuscript technically sound, and do the data support the conclusions?

Reviewer #1: Partly

Reviewer #2: Yes

2. Has the statistical analysis been performed appropriately and rigorously? 

Reviewer #1: I Don't Know

Reviewer #2: I Don't Know

3. Have the authors made all data underlying the findings in their manuscript fully available?

Reviewer #1: Yes

Reviewer #2: Yes

4. Is the manuscript presented in an intelligible fashion and written in standard English?

Reviewer #1: Yes

Reviewer #2: Yes

5. Review Comments to the Author

Reviewer #1: Dear Authors

Thank you for your work comparing prosthetic options for treatment in femoral neck fractures. The analysis of cost-effectiveness of these options is important in surgical decision making.

Comments:

Abstract:

L25ff: There are more than three surgical options for femoral neck fractures. As you mention below osteosynthesis is a hypothetical option. Hypothetical because in the mentioned age group it is usually not indicated for displaced fractures. Nevertheless, the starting sentence is one of the most important ones, therefore rephrasing is recommended.

Intorduction:

Good introduction.

Register data show much longer follow up periods than the used references. What are the Japanese register numbers?

L58f: Three primary prosthetic options are generally available. There are a lot more surgical options.

Methods:

L97f: This sentence was already used earlier in the manuscript. Removing here is suggested.

L117ff: Registry data should be used and could improve the model

L130f: The 5-year mortality rate after hip fracture is 40%. maybe shorter periods should be looked at.

Results

Why did you not present the costs of BHA?

L206f: This sentence is discussion. The aware reader should know that already.

Table 3: Why are data of BHA missing? Deltacost and -QALY is confusing as it is not all the time the compared to BHA.

L218: I do not understand the following paragraph. Especially the parts about the figure 3. Maybe you could further clarify these figures down to the base for someone who is not very used in these models.

Discussion:

284ff: Generalized sentence. Not needed.

L287ff: Because the parameters and the costs of the model are specific for Japan, there is no possibility to translate the results to other countries in any way. Nevertheless, translation of the costs into pounds and dollars is interesting for the international reader. Line 292 until the end of the paragraph is not supported by the data provided!

302ff: I disagree. The basic data are from Japan. Generalizability to other countries is therefore not given! See also comment above.

307: The study has some limitations not had.

309f: Long-term studies are always beneficial. Maybe you could argument here with mortality rate as well.

Additional limitation: The study is not taking into account the state of the patient and decision making in the case of femoral neck fracture. But it is increasing knowledge about cost-effectiveness and therefore further helps the decision process for the patient and his doctor in case of femoral neck fracture.

Conclusion:

The conclusion about DM THA is not as strong as the advantage of SB THA over BHA. This is not represented in the conclusion sentences. Rephrasing is suggested. BHA is not only an alternative if there are contraindications for THA. Expected mortality of the patient is a very important factor and would change your results.

339ff: These statements are not supported by the data provided.

Figure 2

Why is there a line between the points? The datapoints are not fluent. The y axis has to start at 0 so nobody is overestimating the differences.

Figure 3

I do not understand these graphics. Please further clarify or reduce to important values. Dislocation risk of DM-THA is lower than BHA?

Figure 4

Calculation of willingness to pay is not mentioned in the methods. Could you add this? Why is the willingness to pay of BHA decreasing? This needs discussion. Especially in accordance to mortality rate of old patients and femoral neck fractures.

Figure S1 and S2: All shortenings need to be clarified in the figure description.

Reviewer #2: Thank you for the opportunity to review this study. This study conducted a full economic evaluation of bipolar Hemi, single bearing THA, or double mobility THA as treatment options for displaced femoral neck fractures in active older patients in Japan. The authors were able to complete an exhaustive cost effectiveness analysis based on an array of costs, assumptions and postoperative risks, which were drawn from Japanese health data and high quality, published clinical research data. While, as an orthopedic surgeon, I have limited ability to assess the correctness of the economic analysis that was performed, I believe the assumptions that were used are reasonable. The cost effectiveness conclusions make sense and our in line with previous literature and our own experiences in Switzerland. This study provides a valuable and high-quality model that quantitatively helps prove the effectiveness of THA vs BHA in displaced femoral neck fractures. It also supports the use of DM in most patients as it likely provides some additional benefit at very limited additional absolute cost, while acknowledging that some uncertainty remains in this regard, which is also in line with the literature.

My one recommendation would be to shorten the paragraph between lines 287 and 295. It is not particularly difficult for any reader to convert currency to their own local currency and it should be clear to the reader that depending on current currency fluctuations the precise cost effectiveness threshold may at some points be violated slightly.

Further, the limitations of the model are well described and honest. I do not, however, believe that they significantly detract from the validity of the findings in this case. As the authors suggest, more complex models could be developed, but they are unlikely to change or adversely affect the major finding of this study.

Finally, the paper is well-written and requires no revision in this regard.

Overall, this is a well performed study that provides valuable new insights. I thank the authors for their efforts and recommend that the study be accepted for publication with minor revisions. I would perhaps strengthen the conclusions slightly, by making an additional statement that in effect this study supports the use of THA over BHA in all healthy patients over at least a five year period from a cost-effectiveness perspective. However, overall, I would note that the treatment option most appropriate to each patient individually should be chosen.

6. PLOS authors have the option to publish the peer review history of their article (what does this mean?). If published, this will include your full peer review and any attached files.

Reviewer #1: **Yes: **Samuel Haupt

Reviewer #2: No

---

## [Author Response · Author response to Decision Letter 0]

1 Sep 2024

Dear Reviewer 1: We appreciate your constructive comments and suggestions. We have revised the manuscript in accordance with your comments. 

Dear Reviewer 2: Thank you for your positive evaluation of our study.

---

## [Editor Report · Decision Letter 1]

5 Sep 2024

A cost-effectiveness analysis of three surgical options for treating displaced femoral neck fractures in active older patients in Japan: A full economic evaluation

PONE-D-24-17748R1

Dear Dr. Yokoyama,

We’re pleased to inform you that your manuscript has been judged scientifically suitable for publication and will be formally accepted for publication once it meets all outstanding technical requirements. Following revision, your manuscript is really much better. The authors can be proud: Due to demographic changes your study is getting more attention.

Kind regards,

Hans-Peter Simmen, M.D., Professor of Surgery

Academic Editor

PLOS ONE
---

## [Editor Report · Acceptance letter]

11 Sep 2024

PONE-D-24-17748R1 

PLOS ONE

Dear Dr. Yokoyama, 

I'm pleased to inform you that your manuscript has been deemed suitable for publication in PLOS ONE. Congratulations! Your manuscript is now being handed over to our production team.

Kind regards, 

on behalf of

Dr. Hans-Peter Simmen 

Academic Editor

PLOS ONE